# Targeting of the Fun30 nucleosome remodeller by the Dpb11 scaffold facilitates cell cycle-regulated DNA end resection

Susanne CS Bantele[1], Pedro Ferreira[2], Dalia Gritenaite[1], Dominik Boos[2], Boris Pfander[1]*

[1]DNA Replication and Genome Integrity, Max Planck Institute of Biochemistry, Martinsried, Germany; [2]Centre for Medical Biotechnology, Molecular Genetics II, University Duisburg-Essen, Essen, Germany

**Abstract** DNA double strand breaks (DSBs) can be repaired by either recombination-based or direct ligation-based mechanisms. Pathway choice is made at the level of DNA end resection, a nucleolytic processing step, which primes DSBs for repair by recombination. Resection is thus under cell cycle control, but additionally regulated by chromatin and nucleosome remodellers. Here, we show that both layers of control converge in the regulation of resection by the evolutionarily conserved Fun30/SMARCAD1 remodeller. Budding yeast Fun30 and human SMARCAD1 are cell cycle-regulated by interaction with the DSB-localized scaffold protein Dpb11/ TOPBP1, respectively. In yeast, this protein assembly additionally comprises the 9-1-1 damage sensor, is involved in localizing Fun30 to damaged chromatin, and thus is required for efficient long-range resection of DSBs. Notably, artificial targeting of Fun30 to DSBs is sufficient to bypass the cell cycle regulation of long-range resection, indicating that chromatin remodelling during resection is underlying DSB repair pathway choice.

*For correspondence: bpfander@ biochem.mpg.de

## Introduction

DNA end resection – the nucleolytic digestion of the 5' strands of a DSB – is essential for the initiation of homologous recombination (HR) or related recombination-based mechanisms (reviewed in [*Cejka, 2015*; *Symington, 2014*; *Symington and Gautier, 2011*]). At the same time, resection interferes with ligation-based repair (non-homologous end-joining, NHEJ) and thus is the critical step for repair pathway choice. In mitotically dividing cells, recombination-based repair critically depends on the presence of a sister-chromatid. DSB repair pathway choice and accordingly DNA end resection are therefore highly regulated during the cell cycle: in G1 phase, little resection occurs and NHEJ is therefore favoured. Conversely, in S, G2 and M phase, resection is up-regulated and HR becomes more prevalent (*Cejka, 2015*; *Ira et al., 2004*; *Symington, 2014*; *Symington and Gautier, 2011*).

The nucleases that mediate resection can be subdivided into resection initiation (by Mre11-Rad50-Xrs2 and Sae2 in budding yeast) and long-range resection (by Exo1 or Dna2 with Sgs1-Top3-Rmi1 in budding yeast) pathways (*Cannavo and Cejka, 2014*; *Cejka et al., 2010*; *Mimitou and Symington, 2008*; *Niu et al., 2010*; *Zhu et al., 2008*). So far, Sae2 and Dna2 were shown to be cell cycle - controlled by cyclin-dependent kinase (CDK) phosphorylation in yeast (*Chen et al., 2011*; *Huertas et al., 2008*) and EXO1 in human cells (*Tomimatsu et al., 2014*). Notably, however, a bypass of this control is not sufficient to allow efficient end resection to occur in G1, suggesting that other factors may be involved in the cell cycle control of DNA end resection.

**eLife digest** DNA is continually exposed to chemicals and radiation that cause various forms of DNA damage. One of the most toxic forms of DNA damage is the double strand break, in which both strands of the double helix are broken. These breaks can be mended in two ways: by directly joining the broken ends together, or via a process called homologous recombination. In homologous recombination, a duplicate DNA molecule is used as a template to repair the broken DNA strands. These duplicates only form during particular phases of the cell division cycle, which limits when homologous recombination can take place.

A cell can choose which pathway it uses to repair double strand breaks. However, the first step of homologous recombination – trimming the broken DNA ends in a process called resection – commits a cell to the homologous recombination repair pathway. Cell cycle kinases regulate the cell division cycle and control DNA end resection. This control takes two forms: on the one hand by regulating whether the enzymes that trim the DNA ends are active; and on the other hand by regulating the remodelling of the structure into which DNA is packaged, which is called chromatin. However, it is not known which of these two targets is the limiting factor that determines whether homologous recombination occurs.

A protein called Fun30 that remodels chromatin had been found to be important for promoting resection in budding yeast. Bantele et al. now reveal how the activity of Fun30 is regulated by the cell cycle to limit extensive resection to certain cell cycle phases, where homologous recombination is wanted. During those stages, cell cycle kinases add phosphate groups to Fun30. This enables Fun30 to engage in a protein complex that directs Fun30 to the site of a double strand break to facilitate the resection process.

Bantele et al. also studied artificial versions of Fun30 that were directly fused to components of the protein complex, and so bypassed the controls that limit homologous recombination to particular phases of the cell cycle. These forms of Fun30 enabled resection to take place in phases of the cell cycle where it does not normally occur. This suggests that the remodelling of chromatin by Fun30 is a critical step at which resection is regulated by the cell cycle.

Further experiments showed that the cell cycle regulation of human proteins that are equivalent to Fun30 and another protein in the resection complex is similar to that seen for the yeast proteins. In the future, knowing how these proteins are regulated during resection could help researchers to develop new gene editing methods based on homologous recombination that can be used in cells at any stage of the cell cycle.

Resection is also influenced by the surrounding chromatin. Nucleosomes themselves can be inhibitory to resection enzymes (*Adkins et al., 2013*). Additionally, nucleosome-associated proteins such as budding yeast Rad9 or its functional ortholog in humans, 53BP1, can inhibit resection (*Bunting et al., 2010*; *Lazzaro et al., 2008*; *Trovesi et al., 2011*). Consistent with a barrier function of chromatin and/or Rad9, nucleosome remodellers are recruited to DSBs and promote resection, although the mechanism is poorly understood (*Bennett and Peterson, 2015*; *Bennett et al., 2013*; *Chai et al., 2005*; *Morrison et al., 2004*; *van Attikum et al., 2007*, *2004*). The Swr1-like family remodeller Fun30 (SMARCAD1 in humans) was found to be a critical regulator of resection (*Chen et al., 2012*; *Costelloe et al., 2012*; *Eapen et al., 2012*). Fun30 localizes to chromatin surrounding DSBs and *fun30△* mutant cells show a pronounced defect in long-range resection (*Chen et al., 2012*; *Costelloe et al., 2012*; *Eapen et al., 2012*). Importantly, also SMARCAD1 promotes DNA end resection in human cells, suggesting evolutionary conservation (*Costelloe et al., 2012*). Fun30 itself is a substrate for CDK phosphorylation (*Chen et al., 2012*, *2016*; *Ubersax et al., 2003*), but it has remained unclear by which mechanism Fun30 function is regulated during the cell cycle, how Fun30 is targeted to DNA lesions and if this regulation imposes a bottleneck in the regulation of DNA end resection.

Here, we show that CDK phosphorylation enables Fun30 to form a complex with the phospho-protein-binding scaffold protein Dpb11 and the DNA damage sensor 9-1-1. Formation of this complex is required for proper localization of Fun30 and for efficient long-range resection in M

phase cells. Notably, when we bypass the CDK requirement by directly fusing Fun30 to a subunit of the 9-1-1 complex, we observe long-range resection even in G1–arrested cells. This suggests that the cell cycle regulation of long-range resection can be bypassed solely by artificially targeting Fun30 to DSBs. Finally, we show that also human SMARCAD1 binds to TOPBP1 (human ortholog of Dpb11) in a CDK phosphorylation-dependent manner that involves conserved interaction surfaces, suggesting that the formation of a Fun30-Dpb11 complex is a conserved mechanism of cell cycle regulation that could control DNA end resection and repair pathway choice throughout eukaryotes.

## Results

### Cell cycle-dependent targeting of Fun30 by Dpb11

We identified Fun30 in a two-hybrid screen for interactors of the scaffold protein Dpb11. Dpb11 is a critical regulator of genome stability in budding yeast and as such is found in several distinct protein complexes (*Gritenaite et al., 2014*; *Ohouo et al., 2010, 2013*; *Pfander and Diffley, 2011*; *Puddu et al., 2008*; *Tanaka et al., 2007*; *Zegerman and Diffley, 2007*). Crucial for the formation of these complexes are the two tandem BRCT domains of Dpb11, which are phospho-protein binding modules (*Leung and Glover, 2011*) specific for discrete sets of phosphorylation-dependent interactors. In case of Fun30, the interaction is mediated by BRCT1+2, but not BRCT3+4 (*Figure 1A*, *Figure 1—figure supplement 1*). Using Dpb11 expressed from the strong GPD promoter, we also observed an interaction between Fun30$^{3FLAG}$ and Dpb11 in co-immunoprecipitation (Co-IP) experiments (*Figure 1B*). All Dpb11 complexes characterized so far are cell cycle-regulated (*Gritenaite et al., 2014*; *Ohouo et al., 2013*; *Pfander and Diffley, 2011*; *Tanaka et al., 2007*; *Zegerman and Diffley, 2007*). Thus, we tested the interaction between Dpb11 and Fun30 from cells at different cell cycle stages. We observed that Fun30 interacted with Dpb11 only during late S to M phase, but not in G1 (*Figure 1B–C*, *Figure 1—figure supplement 2*) and this interaction was not influenced by DNA damage (*Figure 1D*).

Since Fun30 is phosphorylated by CDK (*Chen et al., 2012, 2016*; *Ubersax et al., 2003*) and Dpb11 was shown to bind several CDK targets (*Gritenaite et al., 2014*; *Pfander and Diffley, 2011*; *Tanaka et al., 2007*; *Zegerman and Diffley, 2007*), we tested if CDK phosphorylation mediates the Fun30-Dpb11 interaction. Indeed, upon CDK inhibition (using the *cdc28-as1* allele and 1-NMPP1 inhibition) Dpb11 binding to Fun30 was strongly reduced (*Figure 1E*). Accordingly, purified Fun30$^{3FLAG}$ was able to interact with $^{GST}$Dpb11-BRCT1+2 in vitro but only after pre-phosphorylation by CDK (*Figure 1F*), suggesting that the Fun30-Dpb11 interaction as well as its regulation by CDK phosphorylation are direct. Therefore, we sought to identify the CDK phosphorylation sites on Fun30, which are relevant for Dpb11 binding. Interaction mapping using truncated constructs placed the Dpb11 interaction site close to the N-terminus of Fun30 (*Figure 1G*, *Figure 1—figure supplement 3*). Within this region, we identified S20 as well as S28 as critical residues for the Fun30-Dpb11 interaction by two-hybrid and Co-IP binding assays using non-phosphorylatable versions of Fun30 (*Figure 1H–I*, *Figure 1—figure supplement 4*). This suggests that phosphorylation of both residues may create a composite binding surface for Dpb11 BRCT1+2, perhaps similar to the Dpb11-binding surfaces on Rad9 and Sld3 (*Pfander and Diffley, 2011*; *Tanaka et al., 2007*; *Zegerman and Diffley, 2007*; *Zegerman et al., 2010*).

It seemed likely that Dpb11 is involved in targeting Fun30 to DNA lesions. We therefore tested recruitment of *WT* Fun30$^{3FLAG}$ or the corresponding *fun30-SSAA* variant to a site-specific, non-repairable DSB using chromatin immunoprecipitation (ChIP). Indeed, we observed that Fun30-SSAA binding to regions distal of the DSB was reduced compared to *WT* (8–20 kb, *Figure 1J*, *Figure 1—figure supplement 5*), while both versions bound similarly to the immediate vicinity of the DSB (1–3 kb, *Figure 1J*, *Figure 1—figure supplement 5*). This result thus confirms recent observations showing a DSB recruitment defect of *fun30-S20A* and *fun30-S28A* mutants (*Chen et al., 2016*).

Importantly, we could expand these data by generating an experimental tool, which restores the Fun30-Dpb11 interaction in a phosphorylation-independent manner. Since conventional phospho-mimetic mutations failed to promote binding (data not shown), we generated a covalent fusion of the Fun30-SSAA protein directly to Dpb11△N lacking BRCT1+2 (*FUN30-AA-DPB11-276-C* expressed as the only copy of Fun30 from the endogenous promoter, referred to as *FUN30-*

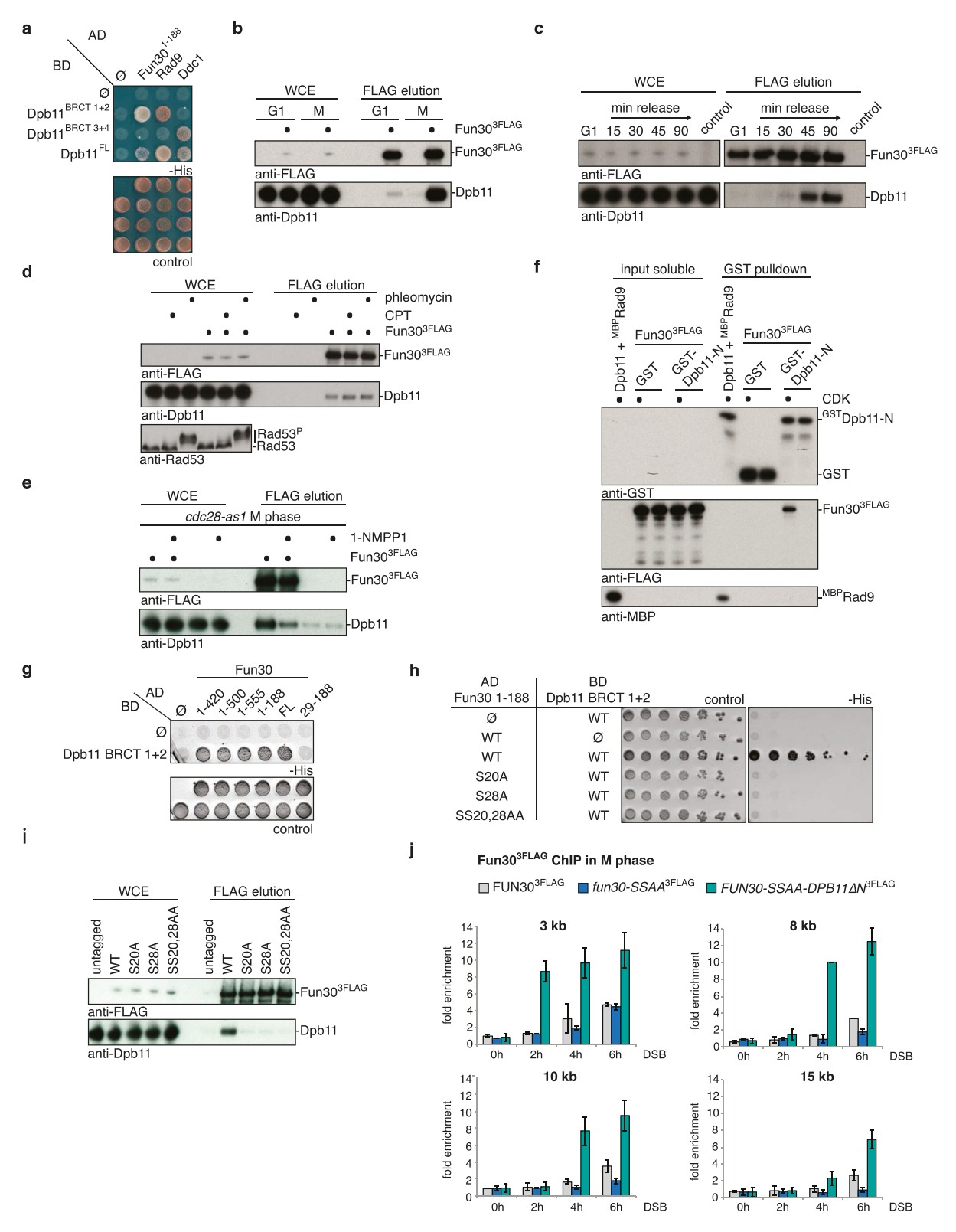

**Figure 1.** Fun30 and Dpb11 interact in a cell cycle- and CDK phosphorylation-dependent manner and this targets Fun30 to DSBs. (a) Two-hybrid assay with GAL4-AD and -BD constructs as indicated reveals a physical interaction between the N-terminal region of Fun30 (aa 1–188) and the BRCT1+2 domain of Dpb11. Rad9 and Ddc1 represent known interactors of BRCT1+2 and BRCT3+4, respectively. (b–e) Characterization of the Fun30-Dpb11 interaction by Fun30^3FLAG Co-IP experiments. Dpb11 was expressed from the strong, constitutive GPD promoter. (b) Fun30^3FLAG specifically binds

*Figure 1 continued on next page*

*Figure 1 continued*

Dpb11 in cells arrested in M but not G1 phase. (c) Fun30[3FLAG] purified from cells synchronously progressing through the cell cycle binds Dpb11 only at 45' and 90' time points corresponding to late S and M phase (*Figure 1—figure supplement 2* for FACS analysis and western analysis of cell cycle progression). (d) No enhancement of the Fun30-Dpb11 interaction by CPT or phleomycin treatment as measured by Fun30[3FLAG] Co-IP. For DNA damage treatment, 50 µM CPT or 50 µg/ml phleomycin were added to asynchronously dividing yeast cells. DNA damage checkpoint activation was measured by Rad53 phosphorylation in IP extracts (lowest blot panel). (e) CDK inhibition using the *cdc28-as1* allele and 1-NMPP1 treatment diminishes the Fun30[3FLAG]-Dpb11 interaction in M phase arrested cells. (f) Purified Fun30 interacts with a BRCT1+2 fragment of Dpb11 in the presence of CDK phosphorylation. Purified Fun30[3FLAG] or the positive control [MBP]Rad9 (*Pfander and Diffley, 2011*) were incubated with a model CDK and ATP before binding to bead-bound [GST]Dpb11 BRCT1+2. (g) Mapping analysis of the two-hybrid interaction between Fun30 and Dpb11 reveals a binding site close to the N-terminus of Fun30. (h–i) Putative CDK sites on Fun30 (S20 and S28) are required for Dpb11 binding. (h) Two-hybrid assay as in (a) but in five-fold serial dilution and with *WT*, *S20A*, *S28A* and *SS20,28AA* variants of Gal4-AD-Fun30[1-188]. (i) Co-IP as in (b) but with mutant variants of Fun30[3FLAG] growing asynchronously. (j) Efficient Fun30 localization to damaged chromatin requires the Dpb11-Fun30 interaction. ChIP of Fun30[3FLAG] to chromatin locations 3, 8, 10 and 15 kb distant of a non-repairable DSB induced at the MAT locus in M phase-arrested cells. *fun30* mutants were expressed from the endogenous promoter as only copy of *FUN30*. The *FUN30-DPB11* fusion contains *fun30-SSAA* and *dpb11△N* mutations. *WT*, *fun30-SSAA* and *FUN30-DPB11 fusion* cells were crosslinked at indicated timepoints after DSB induction. Plotted values represent means from two independent experiments, error bars represent standard deviations.

The following figure supplements are available for figure 1:

**Figure supplement 1.** Expression control of two-hybrid constructs used in *Figure 1A*.

**Figure supplement 2.** Control of the cell cycle states of the experiment in *Figure 1C*.

**Figure supplement 3.** Expression control of two-hybrid constructs used in *Figure 1G*.

**Figure supplement 4.** Expression control of two-hybrid constructs used in *Figure 1H*.

**Figure supplement 5.** Efficient Fun30 localization to damaged chromatin requires the Dpb11-Fun30 interaction.

*DPB11* fusion in the following). Importantly, the fusion protein localized efficiently to damaged chromatin and thus restored the defect of the *fun30-SSAA* mutation (*Figure 1J*, *Figure 1—figure supplement 5*). This finding suggests that the interaction with Dpb11 is indeed involved in targeting Fun30 to DSBs and that the covalent fusion is sufficient to bypass the CDK regulation of Fun30. Notably, DSB recruitment of the Fun30-Dpb11 fusion protein was stronger than Fun30 consistent with the replacement of a transient, PTM-dependent interaction by a covalent interaction. Therefore, we reason that the *FUN30-DPB11* fusion deregulates Fun30 in two ways: first, it uncouples Fun30 from its cell cycle regulation. Second, it leads to enhanced DSB localization to DSBs, thus potentially enhancing Fun30 activity at damaged chromatin.

We also note that the apparently normal recruitment of Fun30-SSAA to the immediate vicinity of the DSB could be explained by an additional CDK phosphorylation-independent, but resection-dependent recruitment mechanism, such as via binding to RPA (*Chen et al., 2012*). Our data thus suggest the existence of two Fun30–targeting mechanisms: one that is Dpb11-dependent and recruits Fun30 to sites of ongoing resection, and a second that is Dpb11-independent and tethers Fun30 to DNA that has already been resected.

## The Fun30-Dpb11 complex is required for efficient long-range resection

We utilized our system of abolishing and constitutively forcing Fun30 binding to Dpb11 in order to investigate the biological function of the Dpb11-dependent Fun30 targeting mechanism and its role in regulating DNA end resection. We measured resection at an HO-induced, non-repairable DSB in M phase-arrested cells using the combined read-out of (a) the accumulation of the ssDNA-binding protein RPA (*Figure 2A*, upper panel) around the DSB by ChIP and (b) the specific DNA loss (occurring due to ssDNA formation, *Figure 2A*, lower panel). Indeed, compared to *WT* cells, *fun30△* mutants showed a pronounced defect in long-range resection, visible by a reduced spreading of both RPA-ChIP and DNA loss, to regions greater than 10 kb away from the break (*Figure 2A*, *Figure 2—figure supplement 1*), confirming previous observations (*Chen et al., 2011, 2012*; *Costelloe et al., 2012*; *Eapen et al., 2012*). Notably, the same defect in long-range resection

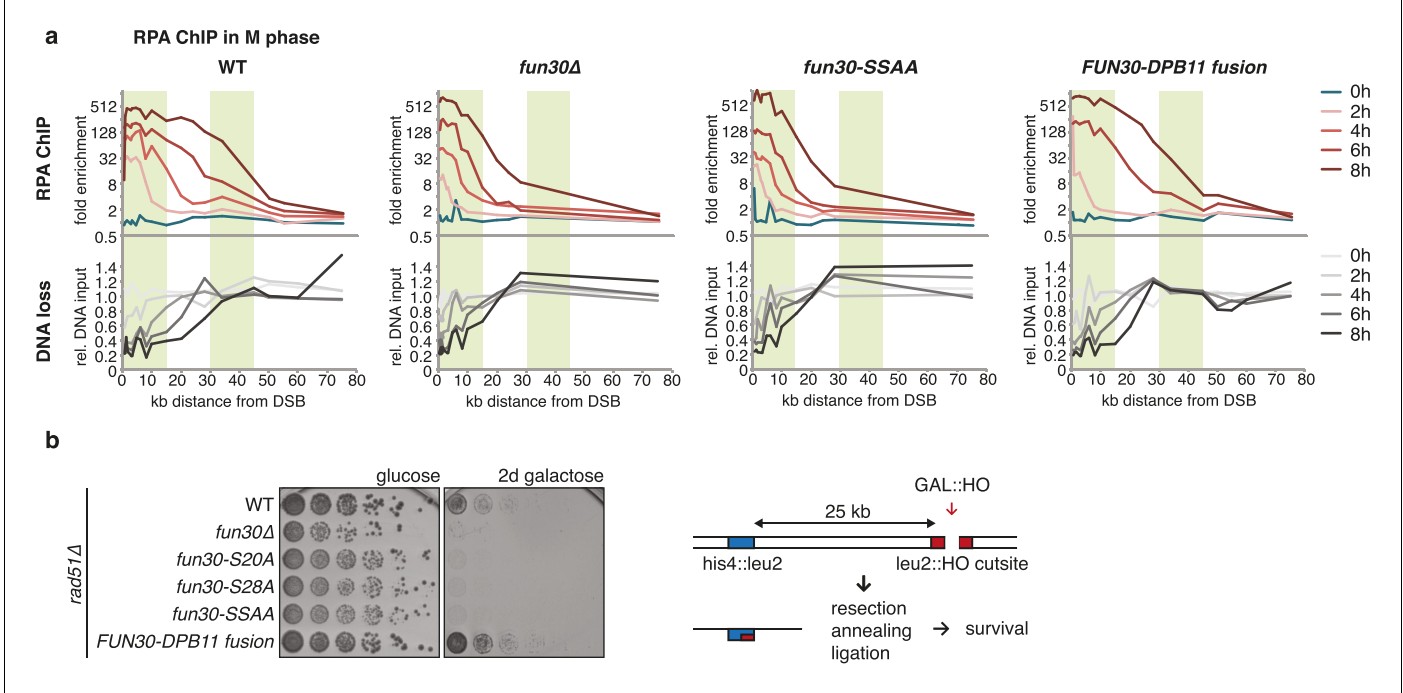

**Figure 2.** The Fun30-Dpb11 complex is required for efficient long-range resection. (**a**) Long-range resection of a DSB is dependent on the Fun30-Dpb11 interaction. A non-repairable DSB at MAT was induced in M phase-arrested *WT*, *fun30Δ*, *fun30-SSAA* and *FUN30-DPB11* fusion strains and DNA end resection measured at indicated times. Upper panel: fold enrichment of a given locus in an RPA ChIP relative to undamaged control loci. Lower panel: DNA loss relative to control loci located in non-damaged chromatin. (**b**) Single-strand annealing (SSA) is dependent on the Fun30-Dpb11 interaction. *FUN30* mutants as indicated were combined with the *rad51Δ* deletion, a DSB at the leu2::HO cutsite was induced by plating cells on galactose. Cells need to resect 25 kb up to the homologous his4::leu2 locus in order to survive by SSA.

The following figure supplement is available for figure 2:

**Figure supplement 1.** The Fun30-Dpb11 interaction is required for efficient long-range resection.

was also observed in the Dpb11-binding deficient *fun30-SSAA* mutant and was fully restored by the *FUN30-DPB11* fusion (*Figure 2A*, *Figure 2—figure supplement 1*). To corroborate these findings, we also analysed resection-dependent DSB repair in a single-strand annealing (SSA) assay, where cellular survival upon an HO-induced DSB in the absence of Rad51 critically depends on the efficient resection of 25 kb of DNA (*Figure 2B*, [*Vaze et al., 2002*]). In fact, Dpb11-binding deficient *fun30* mutants were deficient in SSA-mediated survival and this defect was completely rescued by covalent fusion of Fun30-SSAA to Dpb11 (*Figure 2B*). Thus, the CDK-regulated interaction between Fun30 and Dpb11 is required for efficient long-range resection as well as subsequent resection-coupled repair.

Fun30 participates in chromatin organization in the absence of DNA damage (*Neves-Costa et al., 2009*) and previous studies could therefore not rule out the possibility that the DNA end resection defect of the *fun30Δ* mutant is a consequence of general changes in chromatin organization (*Chen et al., 2012*; *Costelloe et al., 2012*; *Eapen et al., 2012*). However, we found that the *fun30-SSAA* mutant (unlike the *fun30Δ* mutant) did not display any defect in silencing at telomeres or at the silent mating type locus and thus differs from the *fun30Δ* mutant (*Figure 3*). The *fun30-SSAA* mutant thus separates Fun30 functions and the associated resection phenotype of this mutant therefore provides strong support for a direct role of Fun30 and the Fun30-Dpb11 complex during DNA end resection.

Mutants with DNA end resection defects such as *exo1Δ sgs1Δ*, *sae2Δ* or *fun30Δ* are hypersensitive towards the Top1 inhibitor camptothecin (CPT) (*Chen et al., 2012*; *Costelloe et al., 2012*; *Eapen et al., 2012*; *Neves-Costa et al., 2009*) (*Figure 4A–C*), most likely because of repair defects

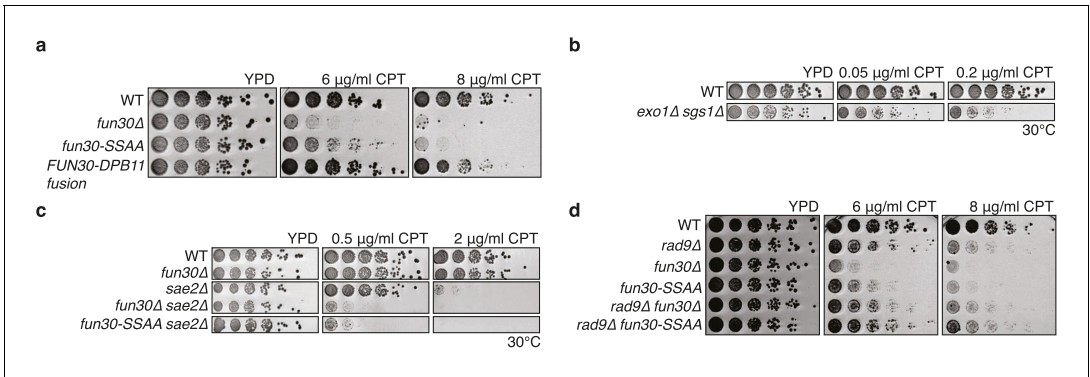

**Figure 3.** The Fun30-Dpb11 interaction is not involved in Fun30-dependent gene silencing at telomeric heterochromatin and a silent mating type locus. The silencing defect of the *fun30Δ* mutant is not recapitulated by the *fun30-SSAA* mutant. Two silencing tester strains were used: the first (upper panels) had URA3 integrated in telomeric heterochromatin at the end of the left arm of chromosome VII, the second (lower panels) had URA3 integrated at the HML silent mating type locus. A silencing defect leads to enhanced growth on –Ura medium and less growth on medium supplemented with 5-FOA (e.g. *fun30Δ*). Shown is a spotting in 5-fold serial dilutions on non-selective medium, medium lacking uracil or containing 5-FOA.

of replication-borne DSBs at CPT-induced Top1 stall sites. Indeed, also the *fun30-SSAA* mutant showed hyper-sensitivity to CPT albeit not as strong as the *fun30* deletion. Importantly, the CPT sensitivity of *fun30-SSAA* was rescued by expressing the covalent *FUN30-DPB11* fusion (**Figure 4A**, **Figure 4—figure supplement 1**), emphasizing again the importance of the Fun30-Dpb11 interaction for DSB repair.

Genetic evidence suggests that Fun30 may promote DNA end resection by antagonizing the resection inhibitor Rad9 (*Chen et al., 2012*). Similar to what has been described for the *fun30Δ* mutant, we observed that the CPT-hypersensitivity of the *fun30-SSAA* was suppressed by an additional *rad9* deletion (**Figure 4D**), suggesting that the Fun30-Dpb11 complex antagonizes Rad9. Interestingly, Rad9 also binds to Dpb11, and Fun30 and Rad9 share the same interaction site on Dpb11 (*Pfander and Diffley, 2011*). While it is currently unknown whether Dpb11-associated Rad9

**Figure 4.** The Fun30-Dpb11 interaction is required for the response towards CPT, as is functional long- and short-range resection. (**a**) The Fun30-Dpb11 interaction is required for the response towards CPT. *WT*, *fun30Δ*, *fun30-SSAA* and *FUN30-DPB11 fusion* were spotted in five-fold serial dilutions on plates containing indicated amounts of CPT and incubated at 37°C for two days. (**b**) A double mutant of *exo1Δ* and *sgs1Δ* is hyper-sensitive to low doses of CPT. Spotting in 5-fold serial dilutions was incubated for two days at 30°C. (**c**) The *fun30Δ/fun30-SSAA* mutants enhance the CPT hyper-sensitivity of *sae2Δ* mutants. Cells were spotted in 5-fold serial dilutions and incubated for two days at 30°C. (**d**) A *rad9Δ* deletion rescues CPT hyper-sensitivity of *fun30Δ* and *fun30-SSAA* mutant alleles.

The following figure supplements are available for figure 4:

**Figure supplement 1.** Mutants of Fun30 show no discernable phenotype upon chronic exposure to HU, MMS or phleomycin.

**Figure supplement 2.** The catalytic activity of Fun30 is required for the suppression of the CPT phenotype in the context of the *FUN30-DPB11* fusion.

(in contrast to nucleosome-associated Rad9) contributes to the inhibition of DNA end resection, the overlapping binding site raised the possibility that Fun30 may interfere with Rad9 function via competition. Therefore, in order to exclude that the *FUN30-DPB11* fusion rescues resection simply by blocking the Rad9-Dpb11 interaction, we inactivated the ATPase activity of Fun30 by a Walker A motif mutation (K603R) in the context of the *FUN30-DPB11* fusion and found the K603R mutant fusion did not restore *WT* resistance to CPT (*Figure 4—figure supplement 2*). Therefore, competition does not explain the effects of the *FUN30-DPB11* fusion and the catalytic activity of Fun30 is required for the resection-promoting function of the Fun30-Dpb11 complex. Overall, these data thus suggest that cell cycle-regulated targeting of Fun30 by Dpb11 is required for efficient DNA end resection.

## The 9-1-1 complex targets Fun30 to DSBs and – as artificial fusion with Fun30 - can be utilized to promote long-range resection in G1

In several organisms, recruitment of Dpb11 and its orthologs to DSBs has been shown to be facilitated by the 9-1-1 complex (*Delacroix et al., 2007*; *Du et al., 2006*; *Furuya et al., 2004*; *Pfander and Diffley, 2011*; *Puddu et al., 2008*), a signalling platform (*Parrilla-Castellar et al., 2004*) which is loaded at DNA damage sites. Given that 9-1-1 interacts with BRCT3+4 of Dpb11 (*Wang and Elledge, 2002*), we tested whether Dpb11 could simultaneously bind to Fun30 and 9-1-1. Indeed, Fun30$^{3FLAG}$ co-precipitated the 9-1-1 subunits Mec3 and Ddc1 and this binding was absent in cells arrested in G1 or in the respective Dpb11 interaction-deficient mutants (*ddc1-T602A* or *fun30-SSAA*; *Figure 5A–B*, *Figure 5—figure supplement 1*). We thus conclude that Fun30, Dpb11 and 9-1-1 can form a ternary complex, which is regulated by the cell cycle stage (model in *Figure 5—figure supplement 2*). Moreover, we observed a reduction of the Fun30 binding in the proximity of a DSB by ChIP, when we interfered either with the 9-1-1-Dpb11 interaction (*ddc1-T602A* mutant) or with the Fun30-Dpb11 interaction (*SLD3-DPB11ΔN* mutant strain, which expresses as only copy of *DPB11* a truncated version of Dpb11 lacking the Fun30 binding site, *Zegerman and Diffley, 2007*), further supporting a role of 9-1-1 in targeting Fun30 to DSBs (*Figure 5C*).

Given that Dpb11 seems to function as an adaptor between Fun30 and 9-1-1, we also generated a covalent fusion of Fun30-SSAA and the 9-1-1 subunit Ddc1 (referred to as *DDC1-FUN30* fusion). Also this fusion rescued the CPT phenotype of the *fun30-SSAA* mutant in a manner that depended on the catalytic activity of Fun30 (*Figure 6A*, *Figure 6—figure supplement 1*). The *DDC1-FUN30* fusion targeted Fun30 even more efficiently to damaged chromatin than the *FUN30-DPB11* fusion and, notably, the corresponding strain was able to survive at very high CPT concentrations, where little growth could be detected even for *WT* cells, indicating that the *DDC1-FUN30* fusion promotes hyper-resistance to CPT (*Figure 6A*). It is thus possible to at least partially overcome the limits of cellular resistance to CPT by providing very efficient targeting of Fun30 to damaged chromatin and uncoupling it from cell cycle control.

DNA end resection is up-regulated in S, G2, M phases of the cell cycle, thus shifting the DSB repair pathway choice from NHEJ to recombination-dependent mechanisms (*Cejka, 2015*; *Symington and Gautier, 2011*). Previous efforts to bypass this regulation have focussed on nucleases (*Huertas et al., 2008*). Thus, we tested if the CDK-regulation of Fun30 may contribute to the cell cycle regulation of DNA end resection or may even be a limiting factor for this regulation. We used the *DDC1-FUN30* fusion, which in contrast to *WT* Fun30 efficiently localized to a DSB in G1 (*Figure 6B*, *Figure 6—figure supplement 3*). This indicates that the fusion may in principle allow Fun30 to act on damaged chromatin in G1, consistent with 9-1-1 being loaded to damaged chromatin in G1 (*Barlow et al., 2008*; *Janke et al., 2010*). Indeed, the *DDC1-FUN30* fusion protein promoted resection in G1 to a significantly larger reach compared to *WT* Fun30, since RPA recruitment could be observed up to 25 kb distance from the DSB (*Figure 6C*, *Figure 6—figure supplement 3*). Notably, the spreading of resection under the *DDC1-FUN30* conditions was even more pronounced than in *WT* cells arrested in M phase (*Figure 6C*, *Figure 6—figure supplement 3*). This effect is thus consistent with the very efficient targeting of Fun30 to DNA damage sites by the *DDC1-FUN30* fusion. This hyperactivation of resection thus indicates that forced tethering of Fun30 to DSB sites is able to bypass the bottleneck that limits long-range resection in G1.

It needs to be pointed out that within the resected region the fold enrichment of RPA recruitment and the extent of DNA loss was not restored to similar levels as observed in M phase (*Figure 6C*).

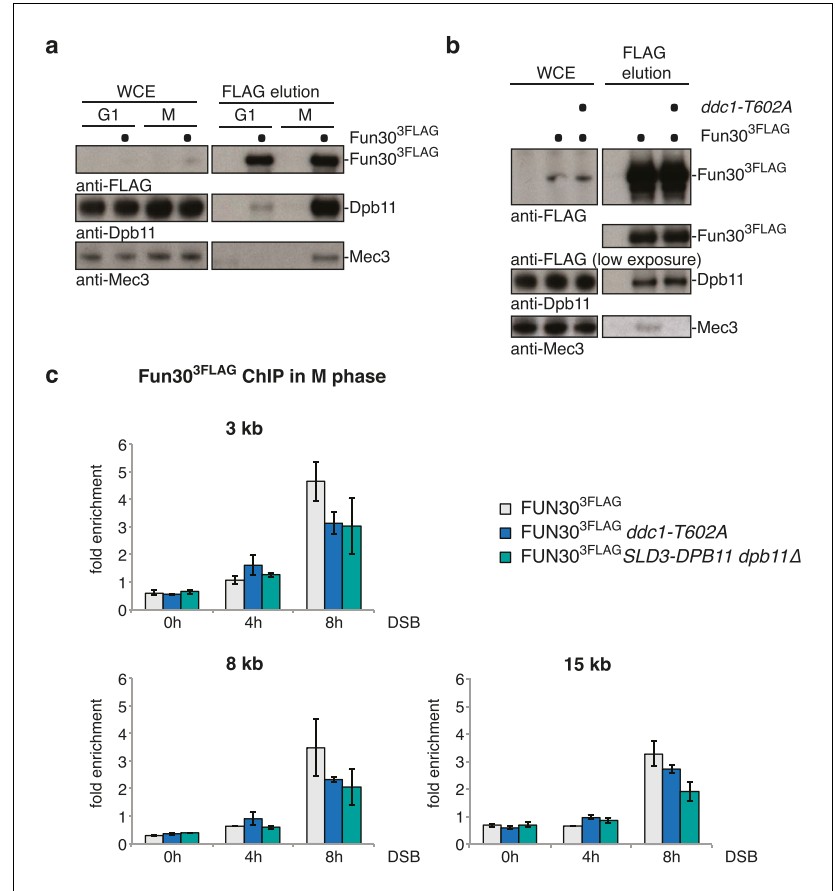

**Figure 5.** The 9-1-1 complex forms a ternary complex with Fun30-Dpb11. (**a**) Fun30, Dpb11 and 9-1-1 form a ternary complex. The 9-1-1 subunit Mec3 interacts with Fun30³ᶠᴸᴬᴳ when purified from M phase cells, where also Dpb11 binds to Fun30. (**b**) The *ddc1-T602A* mutation abolishes binding of Mec3 to Fun30-Dpb11 in Fun30³ᶠᴸᴬᴳ Co-IPs, but leaves the Fun30-Dpb11 interaction intact. (**c**) Mutants disrupting the interaction between 9-1-1 and Dpb11 (*ddc1-T602A*) or Fun30 and Dpb11 (*SLD3-dpb11ΔN,* lacks Fun30 binding site, only copy of Dpb11) impair efficient localization of Fun30 to DSBs in Fun30³ᶠᴸᴬᴳ ChIPs of M phase-arrested cells. Experiment performed as in *Figure 1J*, plotted values represent means of two independent experiments, error bars represent standard deviations.

The following figure supplements are available for figure 5:

**Figure supplement 1.** The interaction between Fun30 and 9-1-1 depends on mutual interactions with Dpb11, suggesting that Dpb11 forms a molecular bridge in the Fun30-Dpb11-9-1-1 complex.

**Figure supplement 2.** Model of the Fun30-Dpb11-9-1-1 association and its regulation.

Moreover, precise ligation of a cut plasmid by NHEJ did not appear to be influenced by the *DDC1-FUN30* fusion (*Figure 7*). These data thus suggest that the overall cell cycle regulation of DNA end resection was not bypassed completely, presumably because other resection proteins and in particular resection initiation are additional targets of cell cycle regulation (*Albuquerque et al., 2008*; *Chen et al., 2011*; *Huertas et al., 2008*; *Pfander and Diffley, 2011*; *Zhang et al., 2009*). We therefore compared G1 resection in the *DDC1-FUN30* strain to another mutant – *sae2-S267E* – that is thought to at least partially bypass the CDK regulation of resection initiation (*Cannavo and Cejka, 2014*; *Huertas, 2010*). Notably, the *sae2-S267E* mutant showed no increase in the reach of resection and only led to a slight increase in the fold enrichment of the RPA ChIP, both in *WT* and *DDC1-FUN30* background (*Figure 8*). Overall, these data are thus consistent with a model, whereby *sae2-S267E* partially bypasses the cell cycle regulation of resection initiation, while the *DDC1-FUN30*

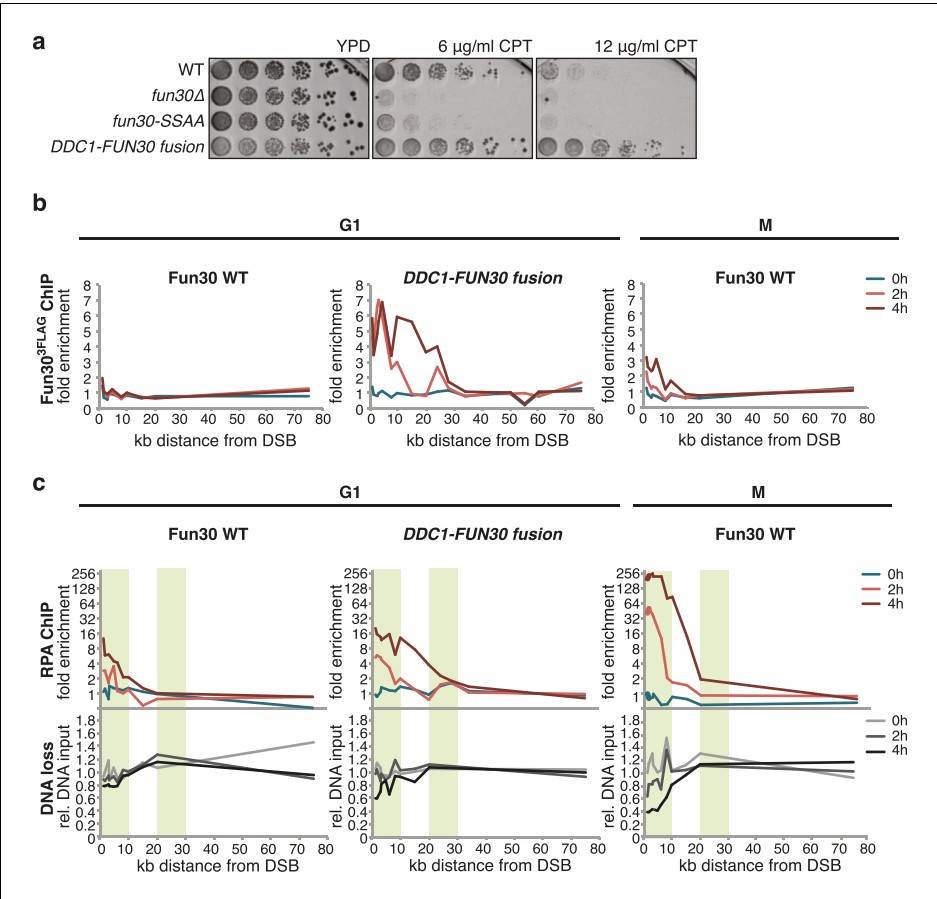

**Figure 6.** A covalent fusion of Fun30 to the 9-1-1 subunit Ddc1 generates a bypass of the cell cycle regulation of long-range resection. (a) The *DDC1-FUN30* fusion confers cellular hyper-resistance to CPT. Spotting of indicated strains as in *Figure 4A*, but using CPT concentrations of up to 12 µg/ml. (b) The *DDC1-FUN30* fusion localizes efficiently to a DSB in G1-arrested cells. Fun30[3FLAG] ChIPs from *WT*, *fun30-SSAA*, *FUN30-DPB11* and *DDC1-FUN30* cells as in *Figure 1J*, but from G1 or M phase-arrested cells. Additional Fun30[3FLAG] ChIP data can be found in *Figure 6—figure supplement 3*. (c) The *DDC1-FUN30* fusion enhances long-range resection in G1-arrested cells. Resection assay as in *Figure 2A*, but with G1 or M phase-arrested cells. Additional resection assay data can be found in *Figure 6—figure supplement 3*.

The following figure supplements are available for figure 6:

**Figure supplement 1.** The *DDC1-FUN30 fusion* rescues the CPT sensitivity of the *fun30△* mutant in a manner that depends on the Fun30 catalytic activity.

**Figure supplement 2.** Flow cytometric analysis of DNA content for experiments shown in *Figure 6B–C* and *Figure 6—figure supplement 3*.

**Figure supplement 3.** The *DDC1-FUN30 fusion* protein efficiently localizes to DSBs and promotes hyper-resection in M phase as well as allowing long-range resection in G1 phase.

fusion bypasses the cell cycle regulation of long-range resection. This highlights that chromatin has a barrier function towards resection and that formation of the Fun30-Dpb11 complex is the limiting step that needs to be up-regulated during recombination-permissive cell cycle phases in order to overcome this barrier.

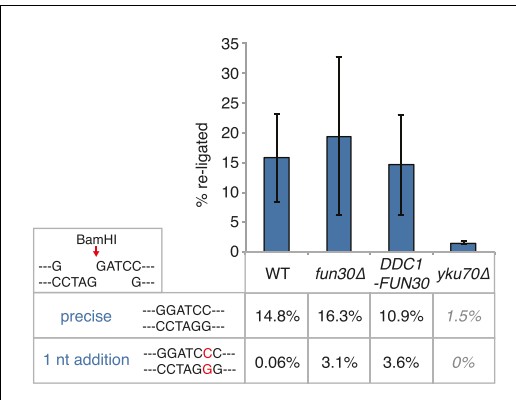

**Figure 7.** The *DDC1-FUN30* fusion does not significantly inhibit non-homologous end-joining (NHEJ). Precise re-ligation of BamHI-cut pRS316 as measured by cell viability on SC-Ura plates and subsequent sequencing of single colonies was dependent on Ku70 but not significantly affected in *DDC1-FUN30* of *fun30Δ* mutant cells. Plotted are values from three independent experiments representing the viability rate of cells on SC-Ura plates relative to the total cell number and the transformation efficiency of the mock-digested plasmid. Error bars represent standard deviations.

## Conservation of Fun30 regulation to human SMARCAD1

Fun30's role in promoting DNA end resection is conserved to its human ortholog SMARCAD1 (*Costelloe et al., 2012*; *Densham et al., 2016*). Strikingly, in an independent screen we identified an N-terminal fragment of SMARCAD1 (aa 55–274) as interactor of TOPBP1 (using a TOPBP1 BRCT0-2 construct), the human ortholog of Dpb11. Furthermore, we found this interaction to require the phospho-protein binding sites of TOPBP1 BRCT1+2 (*Figure 9A*, *Figure 9—figure supplement 1*). We verified the SMARCAD1-TOPBP1 interaction in a pulldown approach using purified GST-TOPBP1-BRCT0/1/2 fragments and in vitro phosphorylation of cell extracts with purified CDK. GFP-SMARCAD1-55-445 bound efficiently to GST-TOPBP1-BRCT0/1/2, but only after addition of active CDK to the cell extract (*Figure 9B*). This suggests that CDK phosphorylation promotes the interaction, similar to the regulation in yeast. We therefore queried for the TOPBP1 interaction site on SMARCAD1 using mutagenesis of CDK consensus motifs. Using this approach, we found that the T71A variant, but none of the other S/TP site mutants tested caused strongly reduced TOPBP1 binding in two-hybrid and Co-IP (*Figure 9C–D*, *Figure 9—figure supplement 2*). These data therefore suggest that SMARCAD1 interacts with TOPBP1 via the CDK-site T71. To our surprise, we observed in two-hybrid experiments that human SMARCAD1 also interacted with yeast Dpb11 and in a manner that was dependent on the T71 phosphorylation site (*Figure 9E*, *Figure 9—figure supplement 3*), despite low sequence conservation. This raised the possibility that SMARCAD1 and FUN30 could also functionally complement each other. Expression of SMARCAD1 from the inducible GAL-promoter lead only to a slight suppression of the CPT sensitivity of a *fun30Δ* strain (data not shown), suggesting that there is an aspect of Fun30 function or regulation that is not recapitulated by SMARCAD1. In contrast, when we generated a SMARCAD1-Fun30 chimera lacking the Dpb11-binding region of Fun30 but containing the TOPBP1-binding region of SMARCAD1 (SMARCAD1-1-300-FUN30-30-C), this chimera was largely able to rescue the CPT sensitivity of the *fun30Δ* mutant (*Figure 9F*, *Figure 9—figure supplement 5*). In contrast, the

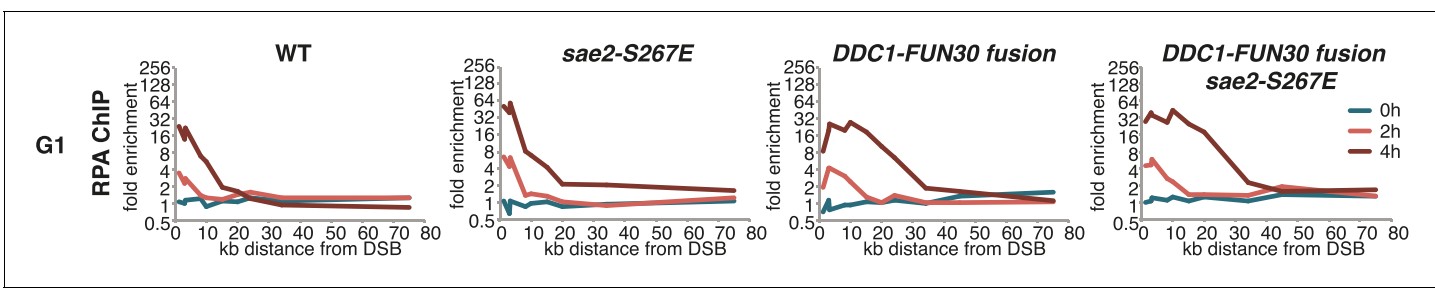

**Figure 8.** The *DDC1-FUN30* fusion specifically enhances long-range resection in G1, while the *sae2-S267E* phospho-mimicry leads to a small increase in resection initiation. The *sae2-S267E* mutant has little effect on the spreading of DNA end resection in G1, but slightly stimulates the RPA fold enrichment in *WT* and the *DDC1-FUN30* fusion mutant. This suggests that *sae2-S267E* in contrast to the *DDC1-FUN30* fusion does not affect long-range resection. DNA end resection in the indicated strains was analysed by RPA ChIP as in *Figure 5C* but with G1 arrested cells.

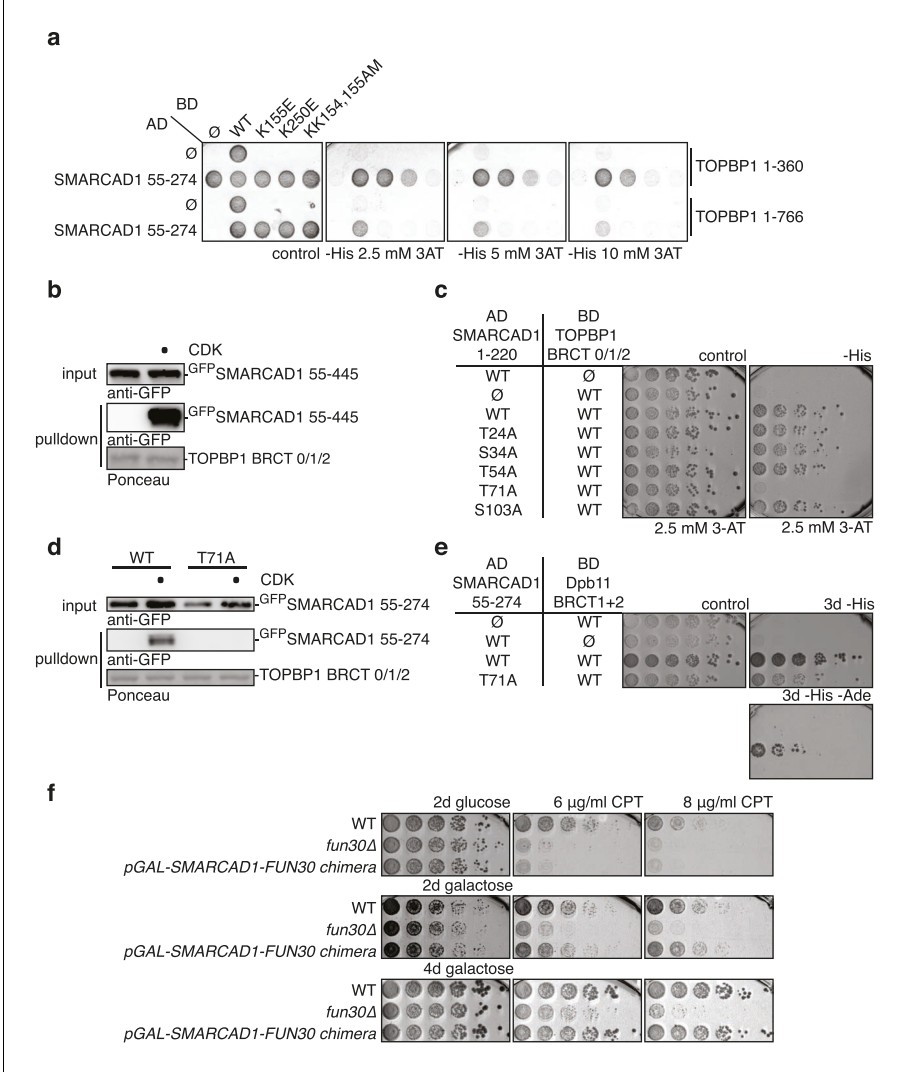

**Figure 9.** Yeast Fun30 and human SMARCAD1 underlie a conserved regulation. (**a**) SMARCAD1 and TOPBP1 interact and their interaction depends on functional phospho-binding pockets in BRCT1 and BRCT2 of TOPBP1. lexA-BD TOPBP1 1–360 (harbouring BRCT0/1/2) or lexA-BD TOPBP1 1–766 (harbouring BRCT0-5) were tested as *WT* versions or as *K155E, KK154,155AM* (affecting BRCT1) or *K250E* (affecting BRCT2) mutant derivatives. Interaction was tested against the Gal4-AD SMARCAD1 55–274. 3AT was added to –His plates to suppress auto-activation and to increase the stringency of the two-hybrid. Two-hybrid interactions with the lexA-BD TOPBP1 1–360 construct were generally stronger compared to lexA-BD TOPBP1 1–766, leading to milder effects of the K155E and K250E single-mutants, particularly at low 3AT concentrations. (**b**) SMARCAD1 interacts with TOPBP1 after CDK phosphorylation. GFPSMARCAD1 (55-445) was bound to a GSTTOPBP1 BRCT0/1/2 construct after phosphorylation with CDK. This CDK-dependent interaction was seen with several N-terminal SMARCAD1 constructs, but not with FL, perhaps due to low expression. (**c–d**) Threonine 71 of SMARCAD1, a putative CDK phosphorylation site, is required for TOPBP1 binding. (**c**) Two-hybrid analysis of ADSMARCAD1 (1-220) and phospho-mutant derivatives to BDTOPBP1 BRCT0/1/2. (**d**) Co-IP as in (**a**), but additionally using a T71A variant of GFPSMARCAD1 (55-274). (**e**) Dpb11 can bind to human SMARCAD1, and T71 is important for the interaction. Two-hybrid analysis as in (**b**), but using a BDDpb11 BRCT1+2 construct. (**f**) A SMARCAD1-Fun30 chimera lacking the Dpb11-binding site of Fun30, but containing the TOPBP1-binding site of SMARCAD1 restores sensitivity to CPT. The SMARCAD1-Fun30 chimera is expressed from the pGAL1-10 promoter and induced by galactose. Spotting on CPT medium as in *Figure 4A*.

The following figure supplements are available for figure 9:

**Figure supplement 1.** The interaction between SMARCAD1 and TOPBP1 depends on functional phospho-binding pockets in BRCT1 and 2 of TOPBP1.

*Figure 9 continued on next page*

*Figure 9 continued*

**Figure supplement 2.** Threonine 71 of SMARCAD1, a putative CDK phosphorylation site, is required for TOPBP1 binding.

**Figure supplement 3.** Dpb11 can bind to human SMARCAD1, and T71 is important for the interaction.

**Figure supplement 4.** A SMARCAD-FUN30 chimera lacking the Dpb11 binding site of Fun30 but containing the putative TOPBP1 binding site of SMARCAD1 restores sensitivity to CPT, while expression of the Fun30 construct lacking the Dpb11 binding site does not.

**Figure supplement 5.** Expression control of the SMARCAD1-FUN30 chimera in *Figure 9F*.

**Figure supplement 6.** Expression control of the SMARCAD1-FUN30 chimera, FUN30-30-C and GFP-FUN30 30-C in *Figure 9—figure supplement 4* .

---

Fun30-30-C fragment alone was unable to provide a rescue and showed reduced protein stability as well (without tag or GFP-tagged, figure *Figure 9—figure supplements 4,6*). This experiment may thus indicate that the TOPBP1-binding region of SMARCAD1 could replace the Dpb11-binding region of Fun30 in vivo. Overall, we therefore conclude from the data in *Figure 9* that the Fun30/SMARCAD1 interaction with Dpb11/TOPBP1, its regulation by CDK phosphorylation and the corresponding interaction surfaces show a remarkable conservation over more than a billion years of eukaryotic evolution.

## Discussion

Our study reveals that the function of Fun30/SMARCAD1 at DSBs is cell cycle-regulated via interaction with Dpb11/TOPBP1. In budding yeast, this interaction seems to facilitate localization of Fun30 to damaged chromatin in a manner that depends on the 9-1-1 complex. Notably, other interactions may also contribute to Fun30 targeting or function, given that Fun30 was shown to interact with RPA and nucleases (*Chen et al., 2012*) and that SMARCAD1 was recently shown to interact with H2A-ubiquitin (*Densham et al., 2016*). Importantly, however, our data suggests that the interaction with Dpb11 and 9-1-1 is essential for the resection function of Fun30 during the cell cycle. In contrast to the Fun30-Dpb11 complex, the other interactions are seemingly cell cycle-independent and future research will need to show whether they are at all critical for Fun30/SMARCAD1 function.

Once recruited to a lesion, Fun30 will then promote the action of the long-range resection machinery by generating resection-permissive chromatin. It seems clear that Fun30 antagonizes the resection inhibitor Rad9 (*Chen et al., 2012*), but different, non-exclusive mechanisms remain possible. For example, Fun30 could directly remove Rad9 from DNA damage sites or render Rad9-containing chromatin resection-permissive, but also could interfere with Rad9 recruitment by changing chromatin composition. We predict that our system of forced targeting Fun30 to damaged chromatin will be useful to discriminate between these possibilities in the future.

It is furthermore possible that a direct competition for Dpb11 binding between Fun30 and Rad9 contributes to the functional antagonism, similar to what has been suggested for Slx4 and Rad9 (*Cussiol et al., 2015*; *Dibitetto et al., 2016*; *Ohouo et al., 2013*). Dpb11 thus interacts with pro- and anti-resection factors, and the same is true for TOPBP1 (*Cescutti et al., 2010*; *Moudry et al., 2016*). It will thus be interesting to figure out in the future, how binding of potential antagonizing factors is balanced.

Overall, our data suggest that at least two layers of cell cycle regulation of DNA end resection can be distinguished. First, nucleases and nuclease-associated factors are substrate for CDK phosphorylation (*Chen et al., 2011*; *Huertas et al., 2008*) and this may directly activate these enzymes, as has for example been shown for the endonuclease activity of MRX-Sae2 (*Cannavo and Cejka, 2014*). Second, chromatin and nucleosome remodellers may be regulated in a way that generates resection-permissive chromatin at damage sites in cell cycle phases when resection is favoured. The Fun30-Dpb11 complex clearly falls in this second category, as does perhaps Rad9/53BP1, the cell

cycle regulation of which we are only beginning to understand (*Cescutti et al., 2010*; *Pfander and Diffley, 2011*). Notably, deregulation of the second layer such as in the experiments with the *DDC1-FUN30* fusion has so far been the most successful strategy to bypass the cell cycle regulation of DNA end resection (*Figure 6*). This emphasizes the importance of resection regulation by its chromatin substrate and suggests that chromatin (more specifically the Fun30 target on chromatin) is the factor that limits long-range resection in G1 phase cells.

Experimentally manipulating DSB repair pathway choice is a key challenge for future research, because it may allow gene targeting in G1/post-mitotic cells, which are currently refractory to this type of approach, since HR is inefficient under these conditions (*Orthwein et al., 2015*). Notwithstanding the overall complexity of DSB repair pathway choice, our results suggest that modification of the DSB-surrounding chromatin by Fun30/SMARCAD1 should be explored further – particularly in higher eukaryotes – as a tool to experimentally channel DSBs into the HR pathway independently of cell cycle stage.

## Materials and methods

### Yeast strains, cell lines and plasmids

All yeast strains used in this study derive from W303 MATa (strains listed in *Supplementary file 1A*) and were constructed using standard methods (*Janke et al., 2004*). Cells were grown in YP glucose or YP raffinose media at 30°C. For sensitivity spottings on camptothecin, plates were incubated at 37°C for 2 days. The inhibitor-sensitive CDK allele *cdc28-as1* (*Bishop et al., 2000*) was inhibited by supplementing 1NM-PP1 (final concentration 1.5 μM) to the medium. Cell cycle synchronization was performed using alpha-factor (5μg/ml) or nocodazole (5μg/ml) for 2–3 hr.

HEK293-T cells were used in mammalian cell culture experiments. Cells were obtained from the cell services facility of CRUKs London Research Institute, authenticated using STR profiling (Promega Mannheim, Germany) and species determination. They were also tested negative for mycoplasma contamination.

For molecular cloning, genes were amplified from yeast genomic DNA and inserted in plasmids using the In-Fusion HD cloning kit (Clontech Saint-Germain-en-Laye, France). For site-directed mutagenesis, a PCR-based protocol with mutagenic oligonucleotides was used. All plasmids used in this study are listed in *Supplementary file 1B*.

### Yeast two-hybrid interaction assays

The yeast two-hybrid analyses of the protein-protein interactions were performed using either the Gal4-based plasmid system (pGAD-C1, pGBD-C1 [*James et al., 1996*]) in PJ69-7a cells, or the lexA-based plasmid system (pBTM116, Clontech Saint-Germain-en-Laye, France) in L40 cells (Invitrogen Schwerte, Germany).

Transformants were spotted in serial (1:5) or single dilution either on SC-Leu-Trp plates (control) or on SC-Leu-Trp-His plates (selection) and grown at 30°C for 2–4 days. For a specific interaction between TOPBP1 1–360 and SMARCAD1 55–247, spotting plates were supplemented with different concentrations of 3-Amino-1,2,4-triazole (3-AT) (2.5–10 mM). To assess the phosphorylation-specific interaction between SMARCAD1 and Dpb11, cells were additionally spotted on SC-Leu-Trp-His-Ade plates.

All experiments (*Figure 1A,G and H*; *Figure 9A,C,E*) were performed in three technical repetitions per biological repetition (spotting of the same yeast cultures on three separate selection plates) and each interaction was observed in several (2-10) independent experiments (a biological replicate corresponds to a fresh transformation of the Y2H expression vectors, raising of the transformed cells and spotting on selective plates).

### Fun30 Co-Immunoprecipitation

Yeast cells were freshly transformed with pUK1 (pAG416 GPD-Dpb11) and grown to log-phase (OD$_{600}$ 0.5) in SC-Ura medium + 2% glucose (YPD) at 30°C. Cells were cell cycle synchronized as describe above, the arrest was controlled by flow cytometry. To release cells from G1 (*Figure 1C*, *Figure 1—figure supplement 2*), BAR1+ cells were synchronized with 5 μg/ml alpha-factor, washed

twice in pre-warmed SC-Ura medium and resuspended in pre-warmed SC-Ura medium supplemented with nocodazole.

For preparation of extracts, 300 OD yeast cells were harvested, washed in ice-cold sorbitol buffer (1 M sorbitol, 25 mM Hepes pH 7.6), and resuspended in 2 ml lysis buffer with protease and phosphatase inhibitors (100 mM Hepes, 200 mM KOAc, 0.1 % NP-40, 10% glycerol, 2 mM $\beta$-mercaptoethanol, 100 nM ocadaic acid, 10 mM NaF, 20 mM $\beta$-glycerophosphate, 400 µM PMSF, 4 µM aprotinin, 4 mM benzamidin, 400 µM leupeptin, 300 µM pepstatin A) and prepared for lysis using a Spex Sample Prep cryo mill. The extracts were cleared by centrifugation and incubated with anti-FLAG agarose resin (Sigma Munich, Germany) for 30 min (4°C, rotation). After six washes with lysis buffer, Fun30-[3FLAG] was eluted twice with 0.5 mg/ml 3X FLAG peptide (Sigma Munich, Germany). The elutions were pooled and proteins were precipitated with TCA prior to analysis on 4–12% NuPAGE gradient gels (Invitrogen Schwerte, Germany) and standard Western blotting.

Yeast in vivo co-immunoprecipitation experiments were not performed in technical replicates. The number of biological replicates (fresh transformation with the GPD-Dpb11 overexpressing plasmid, raising of the cells, lysis and IP) was two or more, with the exception of *Figures 1C* and *5B*, *Figure 5—figure supplement 1*.

## Protein purification

### Purification of bacterially expressed CDK2/cycA$^{\Delta N170}$
CDK/cycA was purified as described previously (*Brown et al., 1995*).

### Purification of Fun30 from S. cerevisiae
YSB784 was grown in 6 L YP medium + 2% raffinose at 30°C until mid-log phase before expression was induced by addition of 2% galactose. After 3 hr of induction, yeast cells were harvested and washed with 1 M Sorbitol + 25 mM HEPES pH 7.6. Cells were resuspended in 20 ml of Lysis Buffer (500 mM NaCl, 100 mM HEPES pH 7.6, 0.1% NP-40, 10% glycerol, 2 mM $\beta$-mercaptoethanol, 400 µM PMSF, 4 µM aprotinin, 4 mM benzamidin, 400 µM leupeptin, 300 µM pepstatin A, 1x complete protease inhibitor cocktail, EDTA-free) and frozen to drops in liquid nitrogen. Cells were lysed using a cryo mill (Spex Sample Prep). The lysate was thawed and cleared by centrifugation. The extract was incubated with 2 ml equilibrated slurry of anti-FLAG M2 resin (Sigma Munich, Germany). After 2 hr of incubation, the resin was washed six times with 15 CV of lysis buffer. Two elution steps were performed by adding 2 ml 0.5 mg/mL 3FLAG peptide in lysis buffer and incubation for 30 min at 4°C. Next, Fun30[3FLAG] was further purified using a 1 ml MonoQ column. To this end, Fun30 was first brought to 100 mM NaCl, bound to the column and eluted on a 100 mM to 1 M salt gradient over 20 CV. Fun30 containing fractions were pooled, snap-frozen and stored at −80°C.

### Purification of GST-Dpb11 1–275 from E.coli
GST-Dpb11-1-275 was purified as described previously (*Pfander and Diffley, 2011*).

### Purification of GST-TOPBP1-1-360 from E.coli
GST-TOPBP1-1-360 was purified as described previously (*Boos et al., 2011*).

## In vitro analysis of protein-protein interactions
In vitro experiments (*Figure 1F*; *Figure 9B,D*) as depicted were performed once, but confirmed in several different experimental setups (*Figure 9B and D*).

## In vitro CDK phosphorylation and pulldown of Fun30
For in vitro pulldown of Fun30 with Dpb11 after in vitro CDK phosphorylation, GST-Dpb11-N or GST (approx. 18 µg per reaction) were immobilized on Sepharose beads for 1 hr at 4°C. The beads were washed twice in lysis buffer (200 mM KOAc, 100 mM HEPES KOH pH 7.6, 10% glycerol, 0.1% NP-40, 2 mM $\beta$-mercaptoethanol) and resuspended in 100 µl lysis buffer per reaction. For phosphorylation of Fun30 and Rad9 (control), 5 µg purified protein per reaction were dialyzed against lysis buffer (4°C) and supplied with 4 mM ATP and 5 mM MgOAc. Buffer or CDK (2.5 µg per reaction) were added and the reactions were incubated for 30 min at 24°C. Then, the pre-bound Dpb11/GST-beads were added to the phosphorylated proteins and incubated for 1 hr at 4°C. The beads were washed

five times in lysis buffer and eluted by boiling in 2x Laemmli buffer. Proteins were separated by SDS-PAGE and analyzed with standard Western Blotting techniques.

## In vitro CDK phosphorylation and pulldown of SMARCAD1

CDK-dependent pulldowns of SMARCAD1-TOPBP1 (*Figure 9B and D*) were carried out as described (*Boos et al., 2011*) for TRESLIN-TOPBP1 with modifications. HEK293T cells were transfected with pCS2-SMARCAD1-55-275 or pCS2-SMARCAD1-55-445 (carrying an N-terminal GFP tag) and native cell lysates were prepared by lysing the cell pellets in 5x lysis buffer (20 mM HEPES pH 7.5, 250 mM KCl, 0.1% Triton X-100, 5 mM $\beta$-mercaptoethanol, 5% glycerol and Complete EDTA-free Protease Inhibitor Cocktail (Roche Mannheim, Germany)). For CDK phosphorylation, the extract was supplemented with 5 mM ATP, 5 mM MgCl$_2$ and approx. 67 ng/µl cycA/CDK2 or buffer (as control) and incubated for 5 min at 25°C. 200 µl of cell extract were incubated with approx. 10 µg immobilized GST-TopBP1-BRCT0/1/2 for 2 hr at 4°C. The beads were washed with lysis buffer and WCE and bound material were analysed by SDS PAGE, Western Blotting and ponceau staining.

## Chromatin immunoprecipitation (ChIP) and qPCR analysis

For chromatin immunoprecipitation of Fun30 and RPA, cells were grown in YP-Raffinose to an OD of 0.5 and - as indicated for the individual experiments- cell cycle arrest was induced. A double-strand break was introduced by inducing the HO endonuclease from the galactose promoter by addition of galactose to the cultures (2% final). 100 ml samples were crosslinked with formaldehyde (final 1%) for 16 min at indicated timepoints and the reaction was quenched with glycine. Cells were harvested by centrifugation, washed in ice-cold PBS and snap-frozen (RPA ChIPs) or directly processed (Fun30-[3FLAG] ChIPs). For lysis, cell pellets were resuspended in 800 µl lysis buffer (50 mM HEPES KOH pH 7.5, 150 mM NaCl, 1 mM EDTA, 1% Triton X-100, 0.1% Na-deoxycolate, 0.1% SDS) and grinded with zirconia beads using a bead beating device. The chromatin was sonified to shear the DNA to a size of 200–500 bp. Subsequently the extracts were cleared by centrifugation, 1% was taken as input sample and 40% were incubated with either anti FLAG M2 magnetic beads (Sigma Munich, Germany) for 2 hr (Fun30-[3FLAG] ChIPs) or 1.5 hr with anti RFA antibody (AS07-214, Agrisera Vännäs, Sweden) followed by 30 min with Dynabeads ProteinA (Invitrogen Schwerte, Germany, for RPA ChIPs). The beads were washed 3x in lysis buffer, 2x in lysis buffer with 500 mM NaCl, 2x in wash buffer (10 mM Tris-Cl pH 8.0, 0.25 M LiCl, 1 mM EDTA, 0.5% NP-40, 0.5% Na-deoxycholate) and 2x in TE pH 8.0. DNA-protein complexes were eluted in 1% SDS, proteins were removed with Proteinase K (3 hr, 42°C) and crosslinks were reversed (8 hr or overnight, 65°C). The DNA was subsequently purified using phenol-chloroform extraction and ethanol precipitation and quantified by quantitative PCR (Roche LightCycler480 System, KAPA SYBR FAST 2x qpCR Master Mix, KAPA Biosystems London, UK) at indicated positions with respect to the DNA double-strand break. As control, 2–3 control regions on other chromosomes were quantified and used for normalization.

Chromatin Immunoprecipitation (ChIP) experiments were generally performed in technical replicates (on the qPCR level, each sample was measured three times). Our experimental design (excluding *Figure 1J* and figure *Figure 1—figure supplement 5*) includes further replicates, which can be considered as technical as well as biological replicates: we took samples at different timepoints after induction of the DNA break, which showed a consistent trend over the experiment. Additionally, we measured signals with 15–20 qPCR primer pairs over the damaged chromosome including three control regions on unaffected chromosomes. Therefore, we plot results from single experiments and timepoints and do not include error bars. Nonetheless, 2–6 repetitions (independent cell growth and crosslinking) were performed for mutants analysed in *Figure 2A*; *Figure 6B–C*; *Figure 2—figure supplement 1*, *Figure 8* with the exception of *Figure 6—figure supplement 3*. The ChIP experiment in *Figure 1J* and figure *Figure 1—figure supplement 5* was performed three times, error bars represent the standard deviation.

## Yeast growth assays

Yeast growth assays (DNA damage sensitivity spottings *Figures 2B* and *4A–D*; *Figure 6A*, *9F*; *Figure 4—figure supplements 1,2*; *Figure 6—figure supplement 1*; *Figure 9—figure supplement 5* and *URA3* silencing assay *Figure 3*) were performed in three technical repetitions per biological repetition (spotting of the same yeast cultures on three separate selection plates), biological

replicates refer to raising of the mutant strains and spotting on plates with indicated conditions. The genotypes were additionally confirmed by comparing several clones of each strain. Each experiment was spotted with three technical replicates.

### Single strand annealing spotting assay

Derivates of the YMV80 strain (*Vaze et al., 2002*) carrying a 25 kb spacer between a galactose-inducible HO cutsite and repair locus with a *rad51△* deletion to prevent DSB repair by gene conversion were grown to stationary phase in YP-Raffinose and spotted in a 5-fold serial dilution on plates containing glucose or galactose. The plates were incubated for 2–3 days at 30°C.

### DNA damage sensitivity spotting assay

Cells were grown to stationary phase in YP-Glucose and spotted in a 5-fold serial dilution on plates containing camptothecin (concentrations as indicated, typically between 4 and 12 µg/ml) or other drugs at the indicated concentrations. The plates were incubated for 2 days at 30°C or 37°C (for camptothecin, if not indicated differently).

### Plasmid re-ligation assay

In order to assay for precise non-homologous end-joining, 40 OD of transformation-competent yeast cells were transformed with 500 ng BamHI-linearized or mock-digested pRS316. Transformed cells were plated in a five-fold serial dilution on SC-Ura and SC-complete agar plates and grown for two days. Clones on plates containing 50–200 clones were counted to calculate the re-ligation rate (ratio of clones from +BamHI –Ura by +BamHI +Ura divided by the equivalent ratio of mock digested plasmid transformations). Each sample was plated in triplicates and the experiment was independently repeated three times. Error bars represent standard deviations.

50–75 clones from the BamHI-digest transformation on –Ura plates were sequenced to analyse the precise NHEJ event. We found that the majority of cells had precisely re-joined the BamHI overhangs, with a subset having added a single G-C basepair at the cutsite (shown as rate in %). *yku70Δ* cells are NHEJ-deficient and showed very low NHEJ rates with exclusively precisely re-joined plasmid sequences, indicating that this number represents the background of uncut plasmid in the reactions.

## Acknowledgements

We thank U Kagerer for technical assistance, J Willmann for early contributions to the project, J Diffley for support of the initial TopBP1 two hybrid screen, J Diffley, J Haber, P Huertas, S Jackson, K-U Reusswig, L Symington for antibodies, plasmid constructs and strains, P Becker, S Gruber, S Jentsch and members of the Jentsch and Pfander labs for stimulating discussion and critical reading of the manuscript. The Fun30-Dpb11 and the SMARCAD1-TOPBP1 interactions were discovered in two-hybrid screens by Hybrigenics, Paris, France. This work was supported by the German Research Council (DFG; project grant PF794/3–1, to BP) and the Max Planck Society (to BP), by the NRW Rueckkehrerprogramm from the MIWF of the state of North Rhine-Westphalia (to DB). SCSB was supported by a Chemiefonds stipend of the FCI. The Pfander lab is associated with the DFG-funded collaborative research cluster SFB1064 Chromatin Dynamics.

## Additional information

### Funding

| Funder | Grant reference number | Author |
| --- | --- | --- |
| Deutsche Forschungsgemeinschaft | Project Grant, PF794/3-1 | Boris Pfander |
| Max-Planck-Gesellschaft | | Boris Pfander |
| Fonds der Chemischen Industrie | Fellowship | Susanne CS Bantele |
| NRW Rueckkehrerprogramm from the state of North-Rhine- | | Dominik Boos |

Westphalia

The funders had no role in study design, data collection and interpretation, or the decision to submit the work for publication.

## Author contributions

SCSB, Conceptualization, Investigation, Methodology, Writing—review and editing, Conceived and designed research, Performed experiments, Analysed the data, Reviewed and edited the manuscript; PF, Investigation, Methodology, Writing—review and editing, Performed human cell culture experiments shown in Fig. 9B+D, Analysed the data, Edited the manuscript; DG, Investigation, Methodology, Writing—review and editing, Performed experiments, Analysed the data, Approved the manuscript; DB, Investigation, Methodology, Writing—review and editing, Identified SMARCAD1-TOPBP1 interaction, Performed human cell culture experiments shown in Fig. 9B+D, Analysed the data and edited the manuscript; BP, Conceptualization, Investigation, Methodology, Writing—original draft, Writing—review and editing, Conceived and designed research, Performed experiments, Analysed the data, Wrote the manuscript

## Author ORCIDs

Boris Pfander, http://orcid.org/0000-0003-2180-5054

## Additional files

### Supplementary files

• Supplementary file 1. Yeast strains used in this study. (A) Table 1 lists all *S.cerevisiae* yeast strains used in this study, their relevant genotypes and the source. (B) Table 2. Plasmids used in this study. Table 2 lists all yeast plasmids and mammalian expression vectors used in this study and their relevant features.

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
