## [Decision Letter]

Thank you for submitting your article "Targeting of the Fun30 nucleosome remodeler by the Dpb11 scaffold facilitates cell cycle-regulated DNA end resection" for consideration by *eLife*. Your article has been favorably evaluated by Jessica Tyler (Senior Editor) and three reviewers, one of whom, Gregory Ira (Reviewer #1), is a member of our Board of Reviewing Editors. The following individual involved in review of your submission has agreed to reveal their identity: Jo Morris (Reviewer #2).

The reviewers have discussed the reviews with one another and the Reviewing Editor has drafted this decision to help you prepare a revised submission.

Summary:

The authors identify a cell cycle-dependent interaction between Fun30 and Dpb11-BRCT domains that depends on CDK-mediated phosphorylation at Fun30 S20 and S28. This complex, together with 9-1-1 is important for long range resection in S-M phase. Fusion of Fun30 to a component of 9-1-1 complex or Dpb11 substitutes the need for Fun30 phosphorylation and even restores resection in G1 blocked cells. Evidence for a similar cell cycle control of resection and interaction between human orthologues (SMARCAD1:TOPBP1) is provided. Together this report uncovers new mechanism of cell cycle regulation of DSB ends resection and proposes a possible strategy to optimize gene targeting in G1/post-mitotic cells. The report is well written, timely and interesting.

Essential revisions:

Although the observations are interesting and important, the collective major comments that need to be addressed prior to publication are:

1) The role of Fun30 phosphorylation by Cdk1 in the interaction with Dpb11 is clear. It is also entirely convincing that Dpb11-Fun30 or Ddc1-Fun30 fusion proteins can substitute the need for the Cdk1 dependent Fun30 phosphorylation. However, the strain carrying any of these fused proteins has several phenotypes that indicate hyperactivity of Fun30. These include better growth/faster recovery upon SSA when compared to WT (Figure 2), surprisingly efficient resection in G1 cells, hyper-resistance to CPT (Figure 5), higher recruitment at DSBs than wild type Fun30. Thus, the major concern is that Fun30 fused to Dpb11 has higher and/or even possibly different activity than the phosphorylated form of the Fun30. Testing phosphomimic mutants would address this question and complement shortcomings of fusion protein analysis. It is occasionally observed that phosphomimic mutants do not restore the activity of protein; if this is the case here the conclusions would need to be toned down. Considering surprisingly efficient resection of Fun30-Dpb11 cells in G1 cells (previously observed only in ku mutants) it would be good to test the efficiency of precise NHEJ in cells carrying Fun30-Dpb11 and Fun30-Ddc1. Work published so far does not indicate any role of yeast Fun30 in the initial resection that is important for the choice between NHEJ and HR. Thus, any role of fused proteins in regulation of NHEJ would point to a nonphysiological function of the fusion protein.

2) The model proposed suggests a hierarchy of recruitment with both Ddc1 and 9-1-1 being upstream of Fun30. Testing Fun30 recruitment to DSBs (by ChIP) in a dpb11 mutant that does not interact with Fun30 or in a ddc1 mutant would greatly support this view. A decreased or no recruitment in these mutants is expected, if the model is correct.

---

## [Author Response]

*Essential revisions:*

*Although the observations are interesting and important, the collective major comments that need to be addressed prior to publication are:*

*1) The role of Fun30 phosphorylation by Cdk1 in the interaction with Dpb11 is clear. It is also entirely convincing that Dpb11-Fun30 or Ddc1-Fun30 fusion proteins can substitute the need for the Cdk1 dependent Fun30 phosphorylation. However, the strain carrying any of these fused proteins has several phenotypes that indicate hyperactivity of Fun30. These include better growth/faster recovery upon SSA when compared to WT (Figure 2), surprisingly efficient resection in G1 cells, hyper-resistance to CPT (Figure 5), higher recruitment at DSBs than wild type Fun30. Thus, the major concern is that Fun30 fused to Dpb11 has higher and/or even possibly different activity than the phosphorylated form of the Fun30. Testing phosphomimic mutants would address this question and complement shortcomings of fusion protein analysis. It is occasionally observed that phosphomimic mutants do not restore the activity of protein; if this is the case here the conclusions would need to be toned down. Considering surprisingly efficient resection of Fun30-Dpb11 cells in G1 cells (previously observed only in ku mutants) it would be good to test the efficiency of precise NHEJ in cells carrying Fun30-Dpb11 and Fun30-Ddc1. Work published so far does not indicate any role of yeast Fun30 in the initial resection that is important for the choice between NHEJ and HR. Thus, any role of fused proteins in regulation of NHEJ would point to a nonphysiological function of the fusion protein.*

We agree with the reviewers that with our artificial protein fusions we achieve two effects. First, we bypass the cell cycle regulation of Fun30 targeting. Second, we generate more efficient Fun30 recruitment to damaged chromatin than would be seen for the WT protein (see Fun30 ChIPs in Figure.6—figure supplement 3). This could lead to Fun30 hyperactivation. In fact, we generated the Ddc1-Fun30 specifically for this effect, because we reasoned that it would target Fun30 to damaged chromatin even more efficiently than the Fun30-Dpb11 fusion. Moreover, in case of the Ddc1-Fun30 fusion we have clear indication that it leads to Fun30 hyperactivation, as was rightly pointed out by the reviewers. In case of the Fun30-Dpb11 fusion we do not have such clear evidence of hyperactivation (the SSA efficiency in Figure 2 is in our opinion similar to WT), even though recruitment to DSBs is more efficient than in WT cells. Nonetheless, also in this case hyperactivation cannot be excluded entirely.

We have now rewritten the manuscript to better account for this effect and apologize if this was not entirely clear in the first version of the paper. Indeed, we are convinced that Fun30 hyperactivation in the Fun30-Ddc1 fusion is critical for allowing long-range resection to proceed efficient in G1. From this we conclude that (1) the cell cycle regulation of other factors, which also contributes to the overall cell cycle regulation of DNA end resection, could potentially be overridden by the hyperactivation of Fun30. Notably we also conclude from this finding that (2) Fun30 activity is most likely limiting to resection in WT cells.

While it is difficult to exclude entirely that Fun30 may have gained a new activity in the context of the fusion, we currently do not have any indication of such a phenomenon. We have thus followed the reviewers’ advice and tested whether NHEJ would be inhibited in the presence of the Ddc1-Fun30 fusion. However, we did not observe any significant decrease in NHEJ rates in the Ddc1-Fun30 fusion, when using a plasmid ligation assay (Figure 7). We thus currently have no evidence for a gain of a nonphysiological function.

Finally, we have also extensively tested conventional phosphomimetic mutations already early on in the project. However – as the reviewers expected and as we have previously observed for many Dpb11-binders – D or E mutations (and in particular the critical DD/EE mutation) in Fun30 failed to restore the interaction with Dpb11 (Figure 10). Phosphomimikry is as such not a tool that can be utilized to study Fun30 regulation.

Author response image 1.Phospho-mimetic mutants of Fun30 S20 and S28 disrupt the interaction with Dpb11.(**a**) Two-hybrid spotting of strains expressing phospho-mimetic versions of ADFun30 1-188 (S20D, S28D, SS20,28DD, S20E, S28E and SS20,28EE respectively) and BDDpb11-N. Cells were grown for three days at 30°C. (**b**) Two-hybrid constructs are detected with anti-Gal4-AD (AD-Fun30 1-188) and with anti-Gal4-BD (for BD-Dpb11-N) antibodies.**DOI:**
http://dx.doi.org/10.7554/eLife.21687.032

*2) The model proposed suggests a hierarchy of recruitment with both Ddc1 and 9-1-1 being upstream of Fun30. Testing Fun30 recruitment to DSBs (by ChIP) in a dpb11 mutant that does not interact with Fun30 or in a ddc1 mutant would greatly support this view. A decreased or no recruitment in these mutants is expected, if the model is correct.*

We thank the reviewers for emphasizing the hierarchical targeting of Fun30 depending on 9-1-1 and Dpb11. In the revised version of our manuscript, we used two mutants (next to our *fun30-SSAA*) to specifically address the effect of mutations in the upstream factors: the 9-1-1 mutant *ddc1-T602A* (deficient in binding to and recruiting Dpb11) and the Dpb11 mutant *sld3-dpb11ΔN* (DPB11 is an essential gene, but the Fun30 interaction site (BRCT1+2) can be truncated if fused to Sld3 (Zegerman et al, Nature, 2007)). As predicted by the model, both mutants indeed lead to a reduction in Fun30 ChIPs to damaged chromatin. As discussed already in the context of Figure 1 (Fun30 ChIP in the *fun30-SSAA* mutant) there is residual recruitment occurring in all mutant backgrounds affecting the 9‐1-1-Dpb11-Fun30 pathway. Most likely this reflects a 9-1-1-Dpb11-independent mode of recruitment. As we speculate in the paper, Fun30 could be targeted to resected DNA via an RPA interaction (Chen et al., Nature, 2012). We reason that the second pathway may depend on resection, which would explain why Fun30 recruitment in the *fun30-SSAA* mutant (which is resection-deficient) is more strongly defective than in the *ddc1-T602A* mutant (which is not resection-deficient, because it also influences Rad9 (our unpublished data). Importantly, however, the *fun30-SSAA* mutant is as defective in resection as the *fun30Δ* mutant. This suggests that, even though other recruitment mechanisms may exist, it is the 9-1-1-Dpb11-Fun30 pathway, which is essential for efficient DNA end resection.